# Cultivar Differentiation and Origin Tracing of *Panax quinquefolius* Using Machine Learning Model-DrivenComparative Metabolomics

**DOI:** 10.3390/foods14081340

**Published:** 2025-04-14

**Authors:** Rongrong Zhou, Yikun Wang, Lanping Zhen, Bingbing Shen, Hongping Long, Luqi Huang

**Affiliations:** 1Hunan Provincial Hospital of Integrated Traditional Chinese and Western Medicine, Hunan Academy of Traditional Chinese Medicine, Changsha 410208, China; 2State Key Laboratory for Quality Ensurance and Sustainable Use of Dao-di Herbs, National Resource Center for Chinese Materia Medica, China Academy of Chinese Medical Sciences, Beijing 100700, China; 3Xiangya School of Pharmaceutical Sciences, Central South University, Changsha 410013, China; 4Institute of Chinese Medicine Resources, Hunan Academy of Chinese Medicine, Changsha 410208, China; 5Center for Medical Research and Innovation, The First Hospital of Hunan University of Chinese Medicine, Changsha 410208, China; 6Hunan University of Chinese Medicine, Changsha 410208, China

**Keywords:** American ginseng, metabolomics, machine learning, cultivated-origin traceability

## Abstract

American ginseng (*Panax quinquefolius* L.) is a rare and valuable plant utilized for medicinal and culinary purposes, with its geographic origin and cultivation significantly affecting its quality and efficacy. However, the metabolic differences between cultivated and wild American ginseng are not well understood. An accurate and reliable method for tracing the origin and evaluating the quality of American ginseng is therefore urgently required. This study introduces a UHPLC-Q/TOF-MS-based comparative metabolomics and machine learning strategy for the rapid identification of wild and cultivated American ginseng. Both principal component analysis and hierarchical cluster analysis revealed distinct metabolic phenotypes between wild and cultivated American ginseng. Furthermore, the integration of univariate and multivariate statistical analyses identified eight differential metabolites in the ESI^+^ mode and three in the ESI^-^ mode, including seven ginsenosides. A potential ginsenosides marker panel was used to construct five machine learning models to assist in diagnosing the metabolic phenotypes of American ginseng. The Random Forest model, based on the eight differential metabolites in the ESI^+^ mode, achieved a 100% classification rate in both test and validation sets for distinguishing between wild and cultivated American ginseng. This study highlights the feasibility and application of our artificial intelligence-driven comparative metabolomics strategy for cultivar identification and geographic tracing of American ginseng, offering new insights into the molecular basis of metabolic variation in cultivated American ginseng.

## 1. Introduction

American ginseng (*Panax quinquefolius* L.), a globally recognized medicinal and dietary plant, is renowned for its significant clinical utility and health-promoting properties [1,2]. Phytochemical research has identified triterpenoid saponins, particularly American ginseng ginsenosides, as its primary active compounds [3]. The *Chinese Pharmacopoeia* attributes to American ginseng benefits such as invigorating qi, nourishing yin, clearing heat, and promoting fluid production. Modern pharmacology has also confirmed its ability to enhance central nervous system function, protect the cardiovascular system, boost immunity, stimulate hemodynamic vitality, and exhibit anti-cancer and anti-aging effects [4,5].

Wild American ginseng is predominantly found in North America, specifically within the United States and Canada, between latitudes of 40 to 50° N and longitudes of 67 to 125° W [6]. However, wild populations are rare and challenging to sustain, necessitating advanced cultivation techniques to ensure a stable supply [6,7]. Distinguishing wild ginseng from cultivated varieties is difficult due to their similar appearance. American ginseng is cultivated in three major regions of the world: the United States, Canada and China [8]. Moreover, different cultivars and origins of American ginseng exhibit variations in quality and clinical efficacy [9], with their metabolic differences remaining largely unexplored. Accurate differentiation between wild, cultivated, and differently American ginseng is crucial for cultivar identification, geographical origin tracking, and quality control.

Metabolomics, an emerging high-throughput analytical technique, is capable of detecting changes in the composition and content of low-molecular-weight compounds (<1500 Da) in biological samples [10]. Metabolites, reflecting both genetic and environmental factors, offer direct physiological insights. Liquid chromatography–mass spectrometry (LC-MS) is widely utilized in metabolomics studies due to its high sensitivity and selectivity [11,12]. This technology has been extensively applied in food and drug analysis, aiding in the discrimination of plant phenotypes and the analysis of metabolic variations [10]. However, the application of metabolomics to distinguish between wild and cultivated American ginseng and to trace their origins remains unexplored.

In order to investigate the effects of growth mode and geographical origin on the chemical composition of *P. quinquefolius*, metabolic differences between wild and cultivated *P*. *quinquefolius* and *P*. *quinquefolius* between different origins were analyzed and compared in this study, and we proposed the use of LC-MS-based metabolomics combined with machine learning to differentiate wild American ginseng from cultivated varieties for the first time. The study identified metabolite differences between American ginsengs and successfully developed an accurate artificial intelligence diagnostic model for multiple classifications, which is of great significance for the identification of American ginseng varieties, the traceability of its planted origin, and the promotion of market regulation.

## 2. Materials and Methods

### 2.1. Materials and Instruments

The Agilent 1290UPLC-6540 Q-TOF chromatography–mass spectrometry system and a ZORBAX Eclipse Plus C18 column (3.0 × 100 mm, 1.8 μm) were procured from Agilent Technologies Inc. (Santa Clara, CA, USA). An ultrasonic instrument was supplied by Shanghai Ke-guide Ultrasonic Instruments Ltd., and an S-23 high-speed bench-top centrifuge was obtained from Hunan Xiangyi Instrument Co., Ltd. (Changsha, China). An AUW120D electronic analytical balance was acquired from Shimadzu (Shanghai, China) Laboratory Equipment Co. (Shanghai, China). Chromatography-grade methanol and mass spectrometry-grade acetonitrile were sourced from Merck Millipore, Germany; distilled water was provided by Watson’s Distilled Water Company in Shenzhen, China; and mass spectrometry-grade ammonium formate was supplied by Thermo Fisher Scientific (China) Co. (Shanghai, China).

A total of 35 batches of American ginseng samples were dried at low temperature of 40 °C and collected, comprising 14 wild specimens from North America, 10 batches of Chinese-cultivated American ginseng, and 11 batches of North American-cultivated American ginseng. These samples were authenticated by Associate Professor Long Hongping of Hunan University of Traditional Chinese Medicine. Each batch was accompanied by a voucher specimen, designated as Nos. 202307001 to 202307035, and deposited for reference at the Herbarium of Hunan University of Traditional Chinese Medicine.

### 2.2. Sample Preparation and Metabolite Extraction for LC-MS

The samples of American ginseng were washed and dried at 40 °C until a constant weight was achieved, then were separated and pulverized into powder, and passed through a 60-mesh sieve. A quantity of 1 g precisely weighed fine powder was extracted with 50 mL water-saturated n-butanol solution in reflux extraction for 90 min. After the extraction process, the mixture was allowed to cool, and the volume was adjusted to compensate for any weight loss by adding additional water-saturated n-butanol. The resulting solution was agitated and subsequently filtered. Subsequently, 25 mL of the filtrate was transferred to an evaporating dish for evaporation under controlled conditions. The residue obtained post-evaporation was re-dissolved in 10 mL of a 50% methanol solution, mixed thoroughly, and filtered to prepare for subsequent analysis.

### 2.3. UHPLC-Q-TOFMS-Based Untargeted Metabolomics Analysis

The chromatographic columns, types of mobile phases, and gradient elution procedures were meticulously evaluated. The optimized liquid chromatography–mass spectrometry (LC-MS) conditions for this study are detailed as follows: a ZORBAX Eclipse Plus C18 column (3.0 × 100 mm, 1.8 μm) from Agilent Technologies was utilized. The mobile phase was composed of an organic solvent, acetonitrile (A), and an aqueous phase (B), which included 0.05 mmol/L formic acid (MS-grade) in water for phase B. The gradient elution program was meticulously designed as follows: from 0 to 5 min, 5–15% A; from 5 to 20 min, 15–25% A; from 20 to 25 min, 25–35% A; from 25 to 35 min, 35–45% A; and from 35 to 40 min, 45–55% A. The flow rate was set at 0.4 mL/min, with an injection volume of 2 μL.

For mass spectrometry detection, electrospray ionization (ESI) was employed for ionization, encompassing both positive (ESI^+^) and negative (ESI^-^) modes. Before analysis, an Agilent standard tuning mixture (ESI-L Low Concentration Tuning Mix, G1969-85000) was applied for accurate mass calibration. The primary MS scan range was set from *m*/*z* 100 to 1700, with nitrogen as the desolvation gas at a temperature of 325 °C and a flow rate of 6.8 L/min. The sheath gas temperature was maintained at 350 °C, the capillary voltage was set at 4.0 kV, and the fragment voltage was 150 V. Targeted MS/MS analysis was conducted by scanning differential metabolites in a targeted mode.

### 2.4. LC-MS-Based Data Processing and Multivariate Data Analysis for Screening Differential Metabolites Between Wild and Cultivated American Ginseng

The raw LC-MS data preprocessing was conducted using MassHunter Profinder 8 software, which encompassed peak extraction, identification, matching, alignment, and normalization to generate the metabolite expression matrix. Subsequently, the metabolite content information was analyzed using both univariate and multivariate statistical methods. Univariate statistics were executed utilizing the *t*-test.

Hierarchical cluster analysis and principal component analysis (PCA) were employed for differentiating metabolic phenotypes among various American ginseng samples. Orthogonal partial least squares discriminant analysis (OPLS-DA) was applied to screen for differential components among different American ginseng varieties. The OPLS-DA model’s performance was evaluated using R2, Q2, and cross-validation tests. Differential metabolites among the American ginseng varieties were identified based on the criteria of VIP > 1 in the OPLS-DA model and a *p*-value < 0.05 in the *t*-test.

Ultimately, the differential metabolites among the American ginseng samples were identified by precise molecular weight determination of molecular ion peaks and secondary mass spectral fragmentation analysis. Key metabolites were confirmed by comparison with authentic standards and mass spectral databases.

### 2.5. Differential Metabolites-Based Machine Learning Diagnostic Model for Differing Wild and Cultivated American Ginseng

All machine learning model development and evaluation were conducted using Python version 3.11.4. The performance of various classifiers, including Gaussian Bayes, logistic regression, neural networks, Random Forests, and Support Vector Machines, was assessed for classifying American ginseng based on statistically significant differential metabolites. The American ginseng samples were partitioned into a training set, comprising 25 samples, and a test set, comprising 10 samples, at a ratio of 5:2. A 10-fold cross-validation method was implemented to verify the stability of the algorithms. Model performance was evaluated using precision, recall, F1-score, and confusion matrix metrics.

## 3. Results

American ginseng is extensively utilized across the pharmaceutical, food, nutraceutical, and cosmetic sectors due to its beneficial medicinal and nutritional properties. The cultivation and origin significantly influence the accumulation of bioactive compounds and are pivotal in determining the quality and market value of commercial American ginseng [8]. Thus, establishing a reliable validation method for ascertaining the cultivation method and origin of American ginseng is crucial for preventing fraud and assessing product quality.

### 3.1. Differentiation and Identification of Wild and Cultivated American Ginseng Based on Microscopic Morphology

Morphological identification, the most straightforward and traditional method for food and herbal raw materials [13], facilitates the distinction of anatomical features among different *P. quinquefolius* specimens in this study. It has revealed significant differences between wild and cultivated *P. quinquefolius*, as well as subtle variations among cultivated specimens from various regions. Wild *P. quinquefolius* is characterized by a smaller size, an elongated rhizome, denser transverse striations, and more prominent yellowish-brown dotted resin canals in the bark (Figure 1) compared to cultivated varieties.

Microstructurally, the main roots of both wild and cultivated *P. quinquefolius* are fundamentally similar. However, wild *P. quinquefolius* is distinctive for its resin canals containing yellowish-brown or reddish-brown secretions. In contrast, cultivated *P. quinquefolius* occasionally displays ringed resin canals. Notably, the number of resin canals in cultivated *P. quinquefolius* varies by region (Figure 1). According to previous research, ginsenosides, the primary secondary metabolites of *P. quinquefolius*, are localized in the cortex cambium of the phloem, resin canals, and cambium [14].

### 3.2. Discrimination of Metabolic Subtypes of American Ginsengs in Metabolomics Based on Unsupervised Analysis

Comparative metabolomics, leveraging mass spectrometry, compares metabolite profiles among various samples or conditions, highlighting metabolic disparities in biological systems under diverse physiological and pathological states [15,16]. In agricultural and food science, this technique can optimize cultivation practices and enhance the medicinal plants’ yield and quality by revealing how different cultivation conditions affect their metabolic profiles [17,18]. Our study, centered on comparative metabolomics, underscores the metabolic variation in saponin content, particularly ginsenosides—key active constituents of American ginseng.

American ginseng, valued as a traditional nourishing food and medicine, enjoys widespread popularity globally [4]. Despite its popularity, wild American ginseng faces challenges due to its low yield and limited availability, hindering sustainable utilization. To address this problem, cultivated American ginseng has been developed [19]. Reports indicate that wild American ginseng possesses superior quality and medicinal value compared to its cultivated counterparts. However, differentiating between wild American ginseng, Chinese-cultivated American ginseng, and North American-cultivated American ginseng based on appearance alone is challenging due to their striking similarities, which can be confusing [7,19]. The metabolic composition differences among these varieties remain unclear and warrant further investigation using sophisticated analytical techniques.

Metabolomics data often present challenges such as high dimensionality, noise, and limited sample sizes, requiring rigorous preprocessing and analysis [10]. Machine learning, with its robust analytical and statistical capabilities, is well suited for diagnosing complex datasets, differentiating similar products, analyzing large-scale omics data, and screening biomarkers [20]. In this study, we employed cluster and dimensionality reduction analyses to differentiate metabolic types of wild and cultivated American ginseng. Univariate and multivariate statistical methods were integrated to extract significant features and identify differential metabolites, potentially serving as biomarkers. Utilizing a panel of differential metabolites, we developed a diagnostic model using five machine learning algorithms, with the Random Forest model achieving 100% accuracy in discriminating between wild and cultivated ginseng based on specific metabolites in ESI^+^ and ESI^-^ modes.

The study encompassed the analysis of 35 batches of American ginseng, comprising 14 batches of wild American ginseng (C1–C14), 10 batches of Chinese-cultivated American ginseng (N1–N10), and 11 batches of North American-cultivated American ginseng (W1–W11) (Figure 2). A total of 1981 and 1992 metabolites were detected in the ESI^+^ and ESI^-^ modes, respectively.

Two unsupervised analytical models, principal component analysis (PCA) and hierarchical clustering analysis (HCA), were utilized to discern the chemical phenotypes of wild and cultivated American ginseng based on the metabolite expression matrix. In the ESI+ mode, PCA indicated that quality control samples were centrally clustered in the scatter plot, demonstrating the robustness of the metabolomic assay employed (Figure 2A). Furthermore, PCA delineated three major distinct metabolic phenotypes: wild American ginseng, North American-cultivated American ginseng, and Chinese-cultivated American ginseng (Figure 2B). HCA further revealed that all wild American ginseng samples formed a distinct cluster, while North American- and Chinese-cultivated American ginseng grouped into another chemical subtype (Figure 2C).

Consistent with the ESI^+^ metabolome data, the ESI^-^ mode analysis also revealed distinct metabolic subtypes for wild, North American-cultivated, and Chinese-cultivated American ginseng, as evidenced by both PCA and HCA (Figure 2D–F). These analyses collectively highlighted clear metabolic differences between wild and cultivated American ginseng.

### 3.3. Supervised Analytical Screening of Differential Metabolites in Wild and Cultivated American Ginseng in ESI^+^ Mode

Partial least squares–discriminant analysis (PLS-DA) is a supervised statistical technique that models the relationship between metabolite expression and sample categories, facilitating the prediction of sample classifications [21]. In this study, PLS-DA clearly differentiated the metabolic phenotypes of wild American ginseng, North American-cultivated American ginseng, and Chinese-cultivated American ginseng (Figure 3).

The orthogonal partial least squares–discriminant analysis (OPLS-DA) in ESI+ mode revealed a distinct intergroup separation and intragroup aggregation trend among wild and North American-cultivated American ginseng samples. Two hundred permutation tests confirmed the robustness and reliability of the developed OPLS-DA model. Differential metabolites with varying levels were positioned at the periphery of the scoring plot. Utilizing the screening criteria of a VIP greater than 1 and an absolute pcorr value greater than 0.5, 228 differential metabolites were identified, of which 40 were annotated and characterized. In comparison to wild American ginseng (C), 34 metabolites exhibited reduced levels in North American ginseng (W), while 6 metabolites showed increased levels in North American ginseng (W) (Figure 4).

Similarly, in ESI+ mode, the OPLS-DA model identified 248 distinct metabolites in wild American ginseng (C) and Chinese-cultivated American ginseng (N) samples, with 39 metabolites annotated. In comparison to wild American ginseng (C), the levels of 34 metabolites were reduced in Chinese-cultivated American ginseng (N), while 5 metabolites were elevated (Figure 5).

In the comparison between the two cultivated-American ginseng types, the OPLS-DA model detected 107 unique metabolites between North American-cultivated American ginseng (W) and Chinese-cultivated American ginseng (N), with 11 metabolites identified and annotated. Relative to North American-cultivated American ginseng, eight metabolites were less abundant in Chinese-cultivated American ginseng (N), and three were more abundant (Figure 6).

From the metabolic comparisons among the three groups, eight differential metabolites were significantly distinct across wild American ginseng, North American-cultivated American ginseng, and Chinese-cultivated American ginseng. These included six ginsenosides (Ginsenoside Rg6, Ginsenoside Rg2, Ginsenoside F1, Ginsenoside Rk2, Ginsenoside Rk2 iso, and Ginsenoside Rg10) and two oleanocarpane saponins (oleanolic acid-28-O-beta-D-glucopyranoside and 3,11-dioxo-beta-caryophyllene). The six ginsenosides were particularly abundant in wild American ginseng. Furthermore, the expression levels of Ginsenoside Rk2, Ginsenoside Rk2 iso, and Ginsenoside Rg6 decreased in the following order: wild American ginseng > Chinese-cultivated American ginseng > North American-cultivated American ginseng (Figure 7A–C).

### 3.4. Supervised Analytical Screening of Differential Metabolites Between Wild and Cultivated American Ginseng in ESI^-^ Mode

Orthogonal partial least squares–discriminant analysis (OPLS-DA) demonstrated clear intergroup separation and intragroup aggregation among wild American ginseng (C) and Chinese-cultivated American ginseng samples (N). A 200-permutation test confirmed the robustness and reliability of the developed OPLS-DA model. Differential metabolites exhibiting variable levels were observed at the periphery of the scoring plot. A total of 243 differential metabolites were identified in the screened samples of wild American ginseng (C) and Chinese-cultivated American ginseng; of these, 44 metabolites were identified and annotated, with 26 metabolites showing lower levels in Chinese-cultivated American ginseng (N) and 18 higher in Chinese-cultivated American ginseng (Appendix A).

Similarly, between wild American ginseng (C) and North American-cultivated American ginseng (W), 38 differential metabolites were detected, with 24 being more abundant in North American-cultivated American ginseng (W) and 14 in wild American ginseng (C) (Appendix A).

OPLS-DA demonstrated clear intergroup separation and intragroup clustering among North American-cultivated American ginseng (W) and Chinese-cultivated American ginseng samples (N). Two hundred permutation tests confirmed the robustness and reliability of the developed OPLS-DA model. Differential metabolite compounds exhibiting variable levels were observed at the periphery of the scoring plot. A total of 222 unique metabolites were identified between North American-cultivated American ginseng (W) and Chinese-cultivated American ginseng samples, with 40 metabolites identified and annotated. Specifically, 17 metabolites were found to be more abundant in Chinese-cultivated American ginseng (N), while 23 metabolites were more abundant in North American-cultivated American ginseng (Appendix A).

Upon metabolic comparison across the three groups, three differential metabolites were consistently different: two ginsenosides (Rd2 and Rg6) and one sugar (sucrose). Among these groups, ginsenoside Rd2 was most prevalent in wild ginseng, and sucrose was particularly high in North American ginseng. The expression trend for ginsenoside Rd2 was as follows: Wild American ginseng > Chinese-cultivated American ginseng > North American cultivated American ginseng (Figure 8A–C). We noted that wild American ginseng contains a higher saponin content compared to cultivated American ginseng from both North America and China. Similarly, the growing environment and cultivation methods may affect the biosynthesis and production of ginsenoside components in ginseng, American ginseng and *Panax vietnamensis* [22,23], with extreme natural environments being more favorable for saponin production and accumulation [19,24,25].

### 3.5. Differentiation and Origin Tracing of American Ginseng Species Using Five Machine Learning Classifiers Based on Differential Metabolites

The OPLS-DA model is more suitable for comparing or classifying samples between two groups and less suitable for comparing samples among multiple groups [26], on the contrary, so many machine learning models are easy to compare and classify samples among multiple groups, which contributes to discrimination of multiple groups. Machine learning can extract key information about the characteristics of herbs, food and agricultural products that are closely related to their origin, and establish discrimination and differentiation modeling to more accurately identify and trace their origins, which is of great importance to the herbal, food and agriculture industry [27].

Our study included three groups of samples wild cultivated American ginseng, North American-cultivated American ginseng, and Chinese-cultivated American ginseng, and we hope to use machine learning to easily achieve tri-classification discrimination. We utilized expression matrices of eight differential metabolites identified in the ESI^+^ mode to construct various machine learning classifiers aimed at identifying the origins of American ginseng (Figure 9). The training set results indicated that the Random Forest Classifier and the Support Vector Machine both achieved a 100% discrimination accuracy, outperforming the Gaussian Naive Bayes, Logistic Regression, and Multilayer Perceptron Classifier, which had respective accuracies of 90.91%, 72.73%, and 63.63%. In the validation set, the Random Forest Classifier and the Support Vector Machine maintained their lead with 100% accuracy, followed by the Gaussian Naive Bayes at 90.91%, Logistic Regression at 72.73%, and the Multilayer Perceptron Classifier at 63.63%. Misclassification was observed with the Gaussian Naive Bayes, Logistic Regression, and Multilayer Perceptron Classifier algorithms. The Random Forest Classifier and the Support Vector Machine demonstrated the highest diagnostic performance for American ginseng (Figure 9), suggesting that the integration of metabolomics and machine learning can effectively identify American ginseng from different origins.

Furthermore, using the expression matrix of three differential metabolites from the ESI- mode, we established multiple machine learning classifiers to diagnose the origins of American ginseng. The training set accuracy was highest for the Random Forest Classifier (100.0%), followed by Logistic Regression Classifier and Gaussian Naive Bayes Classifier (both at 95.83%), Multilayer Perceptron Classifier (87.5%), and Support Vector Machine (70.83%). In the test set, the Random Forest Classifier, Logistic Regression, and Gaussian Naive Bayes Classifier all achieved 100% accuracy, with the Multilayer Perceptron Classifier at 90.91% and the Support Vector Machine at 63.64%. Misclassification issues were noted with all algorithms except the Random Forest Classifier. The Random Forest Classifier’s superior diagnostic performance for American ginseng classification is highlighted (Figure 10), reinforcing the utility of combining metabolomics and machine learning for origin tracing of American ginseng.

In our study, we compared the ability of multiple machine models to discriminate the origin of American ginseng, and found that Random Forest exhibited high accuracy for the origin tracing of American ginseng by integrating multiple decision trees. In addition, Random Forest also shows strong resistance to overfitting; due to the different training data and feature sets of each tree, the model has a strong generalization ability, which can effectively allow avoidance of the overfitting problem that easily occurs with a single decision tree.

It is necessary to acknowledge the limitations of this work. Due to the difficulty of collecting samples, we did our best to collect samples from 35 batches. Based on these 35 batches of samples, we conducted a machine learning exploration of the origin traceability of American ginseng. In addition, due to the slow accumulation of material data, the small-sample-size problem will still exist for a long time in food analysis and herbal analysis; exploratory analysis of machine learning models based on small samples (<100) for and food traceabilityis acceptable [28,29]. A future study with a larger cohort size is needed for tracing the origin of American ginseng.

## 4. Conclusions

This study introduces a novel approach by integrating metabolomics with machine learning to differentiate between wild and cultivated American ginseng for the first time. We identified differential metabolites in American ginseng and developed an accurate multiclass diagnostic model. Specifically, eight ESI^+^ and three ESI^-^ differential metabolites were discerned by orthogonal partial least squares–discriminant analysis (OPLS-DA), facilitating the distinction among the three types of American ginseng. Notably, seven ginsenosides were identified as key markers, primarily for differentiating between cultivated and wild American ginseng. Various machine learning models, predicated on these differential metabolites, can reliably distinguish between wild and cultivated American ginseng, with the Random Forest model achieving a perfect 100% discrimination rate in both test and validation sets. This study offers an accurate and efficient method for identifying the origin and metabolic variances of American ginseng, beneficial for preserving market integrity and ensuring quality control. However, due to the scarcity of wild American ginseng, our sample collection was limited. Future studies should aim to expand the sample size, ensuring representation from key cultivation regions to enhance the model’s applicability and generalizability.

## Figures and Tables

**Figure 1 foods-14-01340-f001:**
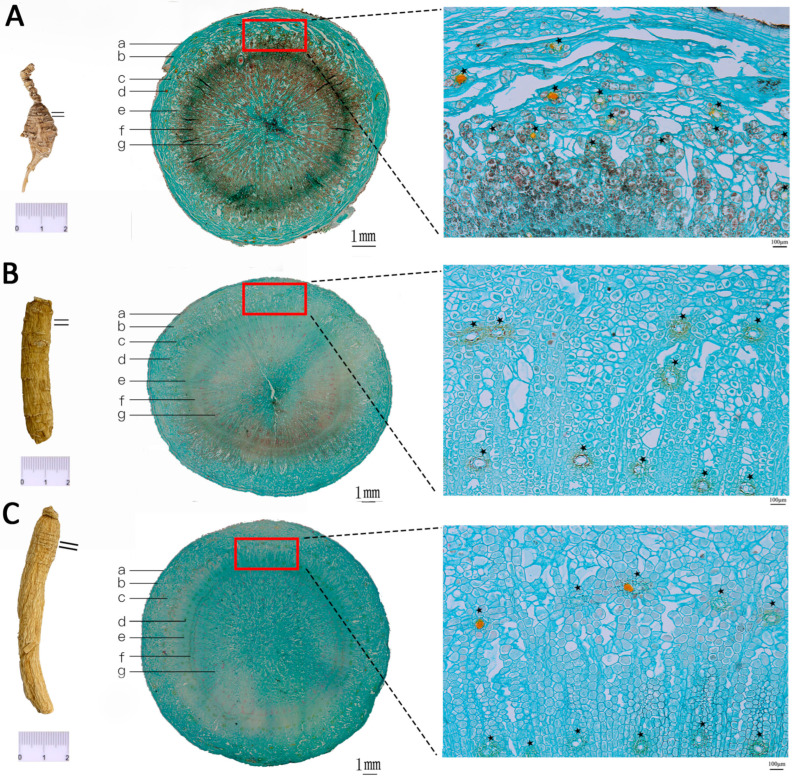
Micrographs of the transverse sections of American ginseng. (**A**) Wild American ginseng; (**B**) North American-cultivated American ginseng; (**C**) Chinese-cultivated American ginseng; a. phellem; b. phelloderm; c. cortex; d. resin canal; e. phloem; f. cambium; g. xylem; ★ resin canal.

**Figure 2 foods-14-01340-f002:**
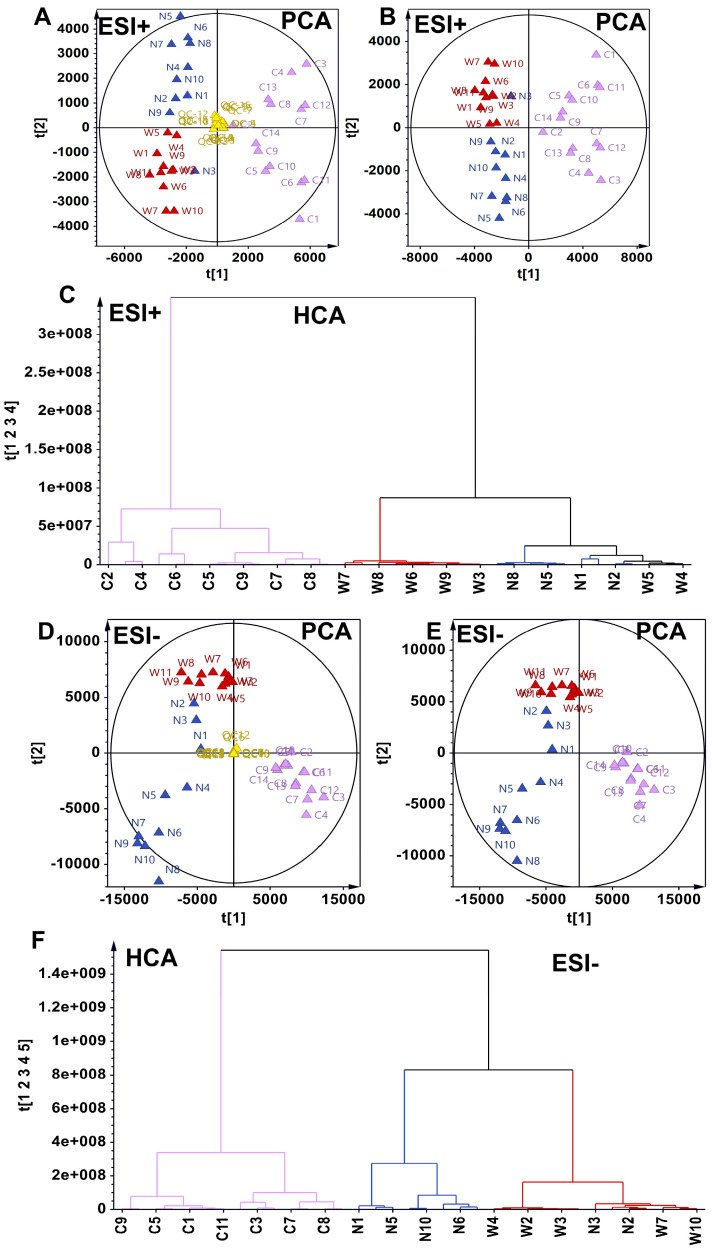
Distinctive metabolic phenotypes in American ginseng as determined by principal component analysis (PCA) and hierarchical clustering analysis in ESI^+^ and ESI^−^ Modes, triangles represent American Ginseng Samples, purple for wild American ginseng (C1–C14), blue for Chinese-cultivated American ginseng (N1–N10) and red for North American-cultivated American ginseng (W1–W11). (**A**) PCA scatter plot including quality control (QC) samples, demonstrating robust clustering and the reliability of the LC-ESI^+^-MS method. (**B**) PCA scatter plot in ESI^+^ mode excluding QC samples. (**C**) Hierarchical clustering analysis highlighting metabolic differences among the three American ginseng types in ESI^+^ mode. (**D**) PCA scatter plot including QC samples, showing robust clustering and the reliability of the LC-ESI^-^-MS method. (**E**) PCA scatter plot in ESI^-^ mode excluding QC samples. (**F**) Hierarchical clustering analysis highlighting metabolic differences among the three American ginseng types in ESI^−^ mode.

**Figure 3 foods-14-01340-f003:**
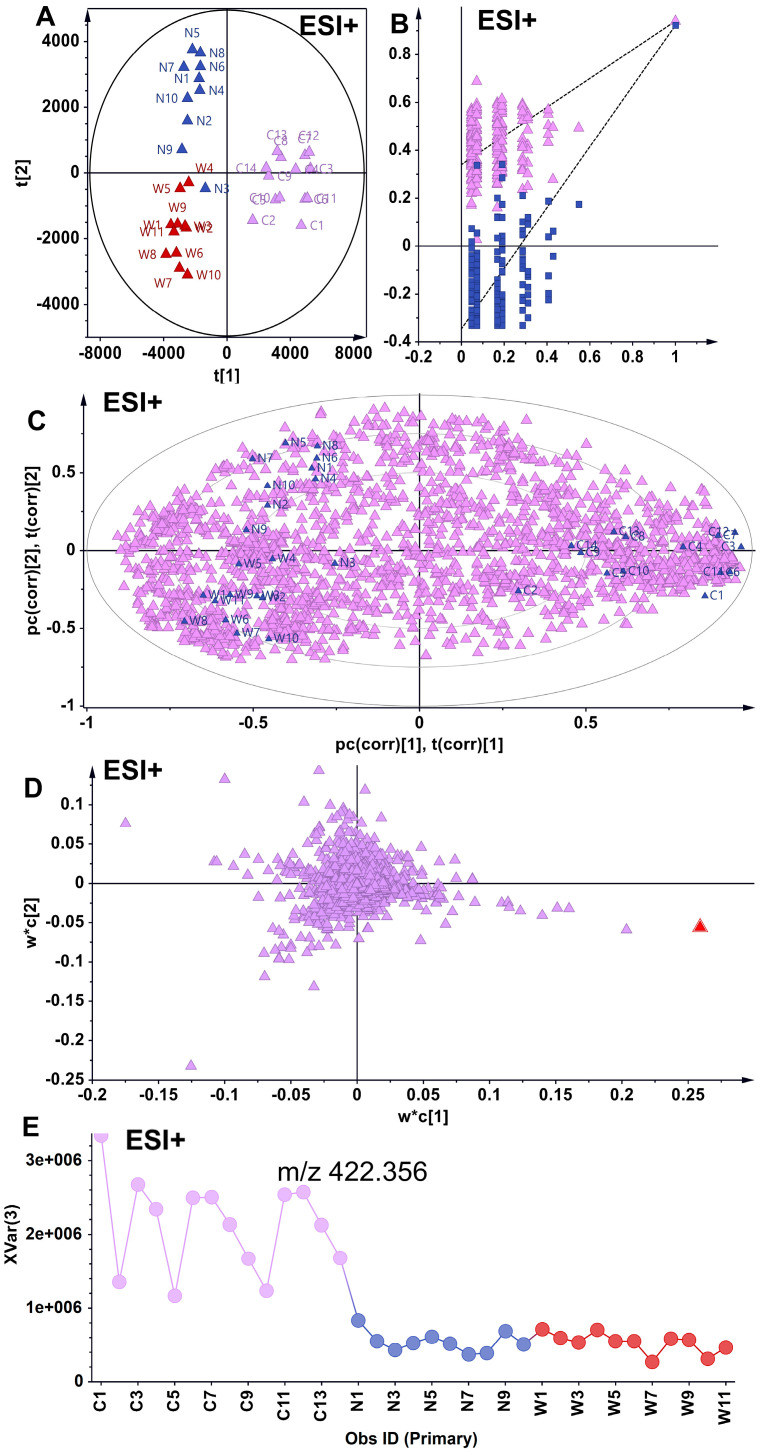
Discrimination of chemotypes among wild, North American-cultivated, and Chinese-cultivated American ginseng using partial least squares–discriminant analysis (OPLS-DA). (**A**) Results depict the segregation and intragroup clustering of wild ginseng and North American-cultivated ginseng samples. C and W represent wild American ginseng (C1–C14) and North American-cultivated American ginseng (W1–W11) (**B**) A 200-permutation test confirms the robustness and reliability of the OPLS-DA model. The purple triangles and blue squares represent R^2^ and Q^2^, respectively. (**C**) Corrplot between group and metabolites. (**D**) Loading plot illustrating differential metabolites, with those exhibiting variable levels localized at the periphery of the scoring plot. Screening was conducted based on criteria where the Variable Importance in Projection (VIP) exceeded 1 and the absolute partial correlation (pcorr) value was greater than 0.5, identifying 228 differential metabolites. (**E**) Metabolite (m/z 422.356) distribution. Dots represent American Ginseng Samples, purple for wild American ginseng (C1–C14), blue for Chinese-cultivated American ginseng (N1–N10) and red for North American-cultivated American ginseng (W1–W11).

**Figure 4 foods-14-01340-f004:**
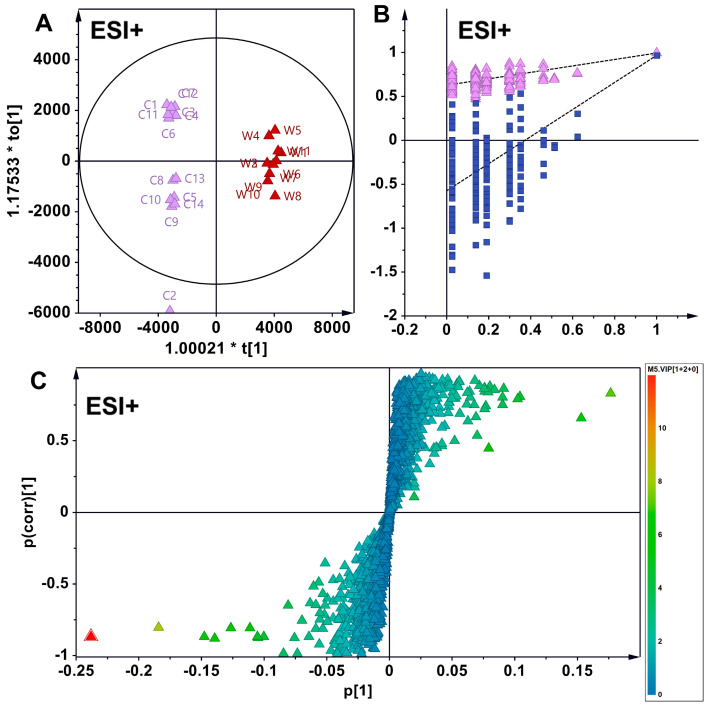
OPLS-DA analysis discriminating wild **(C)** from North American-cultivated American **(W)** ginseng in ESI^+^ mode. (**A**) Scatter plot demonstrating the segregation of wild ginseng and North American-cultivated ginseng samples, with intra-group clustering. C and W represent wild American ginseng (C1–C14) and North American-cultivated American (W1–W10). (**B**) The 200-permutation test confirms the robustness and reliability of the OPLS-DA model. The purple triangles and blue squares represent R^2^ and Q^2^, respectively. The replacement test confirms the model’s stability and reliability. (**C**) Loading plot of differential metabolites, highlighting those with variable content at the periphery. Metabolites were screened based on these criteria: VIP > 1 and |pcorr| > 0.5.

**Figure 5 foods-14-01340-f005:**
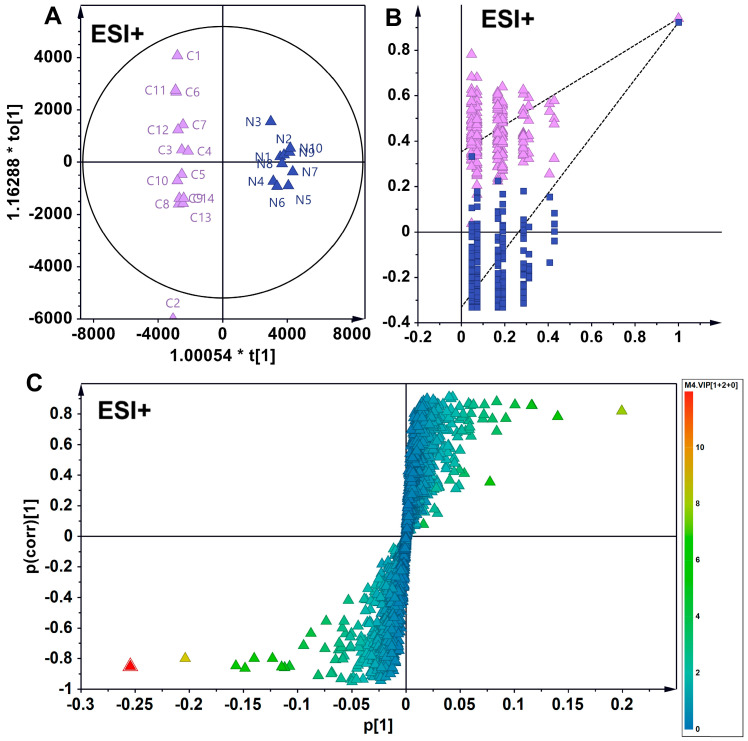
OPLS-DA analysis of differential metabolites in wild (**C**) vs. Chinese-cultivated (N) American ginseng in ESI^+^ mode. (**A**) The analysis reveals distinct segregation and intra-group clustering between wild American ginseng (C1–C14) and Chinese-cultivated American ginseng (N1–N10). (**B**) A 200-permutation test confirms the robustness and reliability of the established OPLS-DA model. The purple triangles and blue squares represent R^2^ and Q^2^, respectively. (**C**) Loading plot depicting differential metabolites, with those exhibiting variable levels positioned at the periphery. Metabolites were identified based on these criteria: VIP > 1 and |pcorr| > 0.5.

**Figure 6 foods-14-01340-f006:**
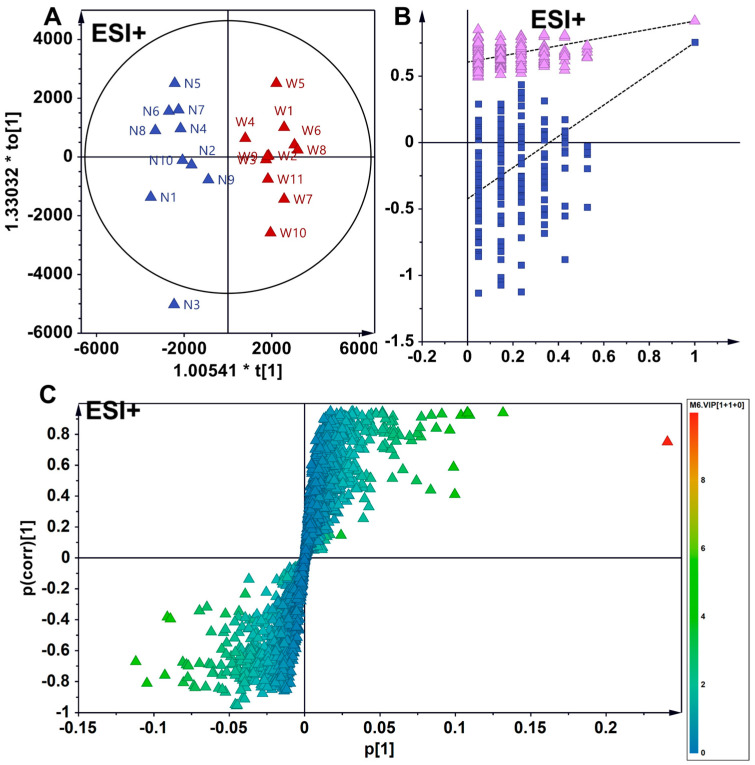
OPLS-DA discrimination of North American (W)- and Chinese (N)-cultivated American ginseng in ESI^+^ mode. (**A**) The analysis illustrates the separation of North American (W) and Chinese (N) cultivated American ginseng samples into distinct groups, with clustering observed within each group. (**B**) The 200-permutation test confirms the robustness and reliability of the OPLS-DA model. (**C**) The loading plot displays differential metabolites, with those showing variable levels located at the periphery. Screening criteria included VIP > 1 and |pcorr| > 0.5.

**Figure 7 foods-14-01340-f007:**
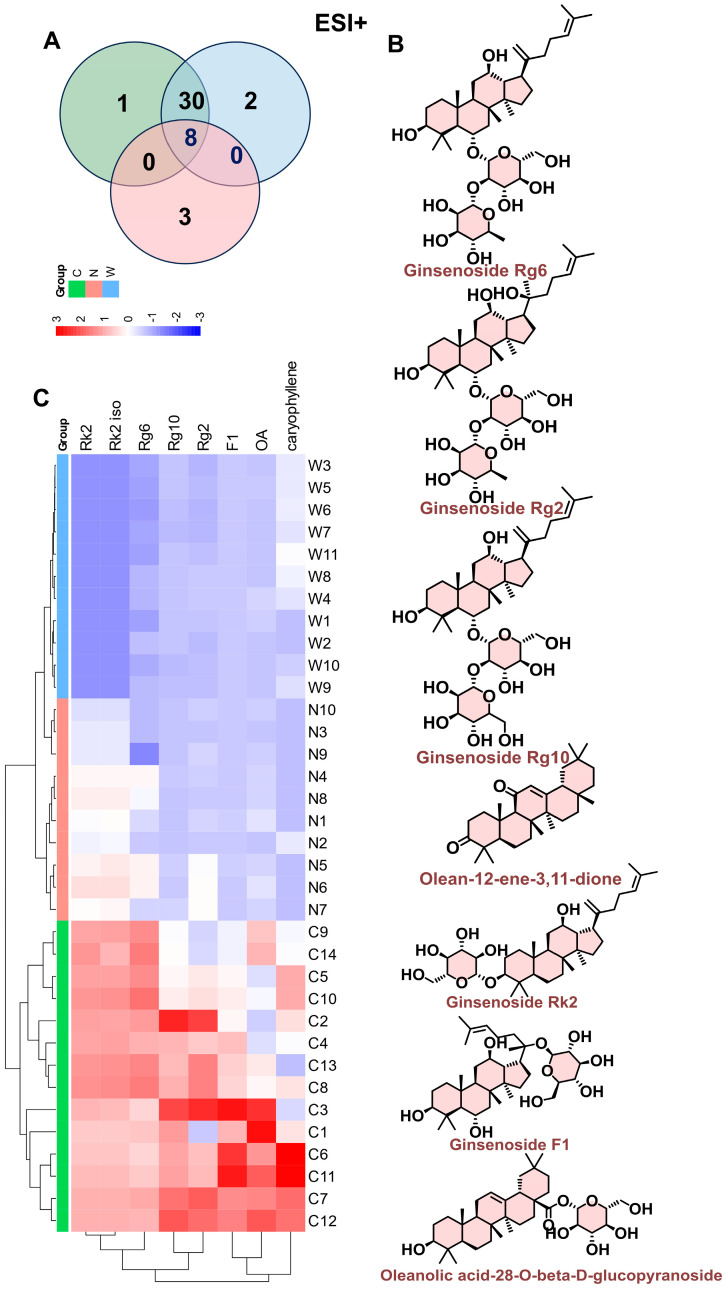
Analysis of ESI^+^ differential metabolites across three groups. (**A**) Comparative analysis of differential metabolites revealed eight distinct compounds across all three groups, comprising six ginsenosides and two oleanane-type triterpenes: Oleanolic acid-28-O-beta-D-glucopyranoside and Olean-12-ene-3,11-dione. The differential ginsenosides identified were Ginsenoside Rg6, Ginsenoside Rg2, Ginsenoside F1, Ginsenoside Rk2, Ginsenoside Rk2 iso, and Ginsenoside Rg10. (**B**) Structural depiction of differential metabolites. (**C**) Expression trends indicated higher abundance of all six metabolites in wild American ginseng compared to North American and Chinese-cultivated ginseng. The trend for Ginsenoside Rk2, Ginsenoside Rk2 iso, and Ginsenoside Rg6 was as follows: wild American ginseng > Chinese-cultivated American ginseng > North American-cultivated American ginseng.

**Figure 8 foods-14-01340-f008:**
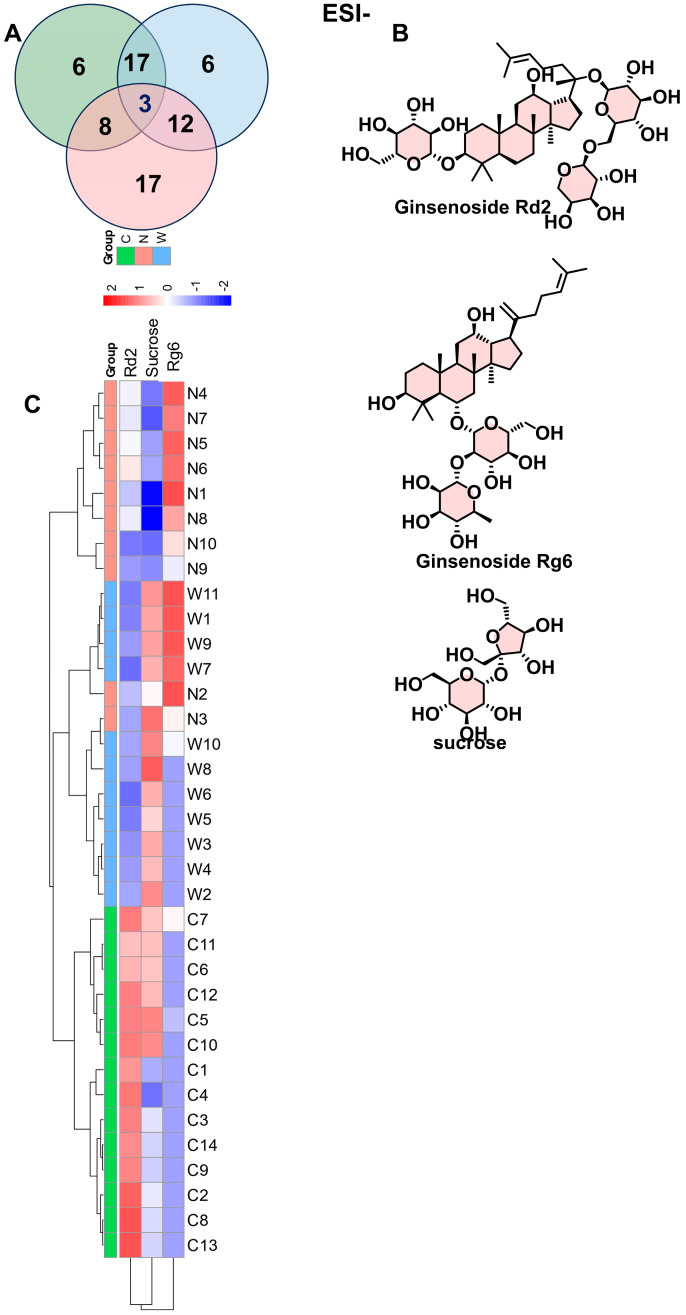
Analysis of ESI^-^ differential metabolites across three groups. (**A**) Comparative analysis of differential metabolites identified three distinct compounds across all three groups, comprising two ginsenosides, Ginsenoside Rd2 and Ginsenoside Rg6, and sucrose. C, N and W represent wild American ginseng (C1–C14), Chinese-cultivated American ginseng (N1–N10), and North American-cultivated American ginseng (W1–W11). (**B**) Structural depiction of differential metabolites. (**C**) Expression trends for Ginsenoside Rd2 indicated the following hierarchy of abundance: wild American ginseng > Chinese-cultivated American ginseng > North American-cultivated American ginseng.

**Figure 9 foods-14-01340-f009:**
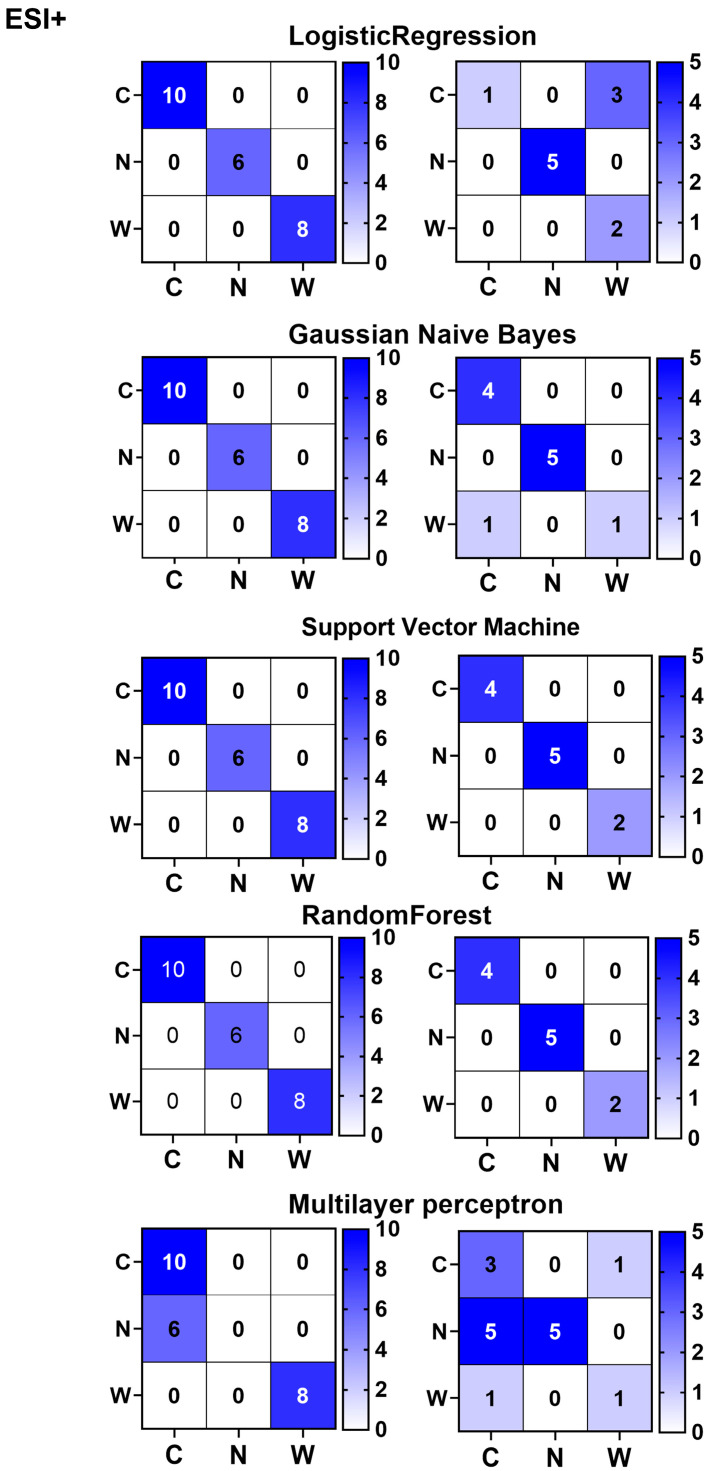
Multiclassifications discriminative models for American Ginseng in ESI^+^ mode using various machine learning approaches. C, N and W represent wild American ginseng (C1–C14), Chinese-cultivated American ginseng (N1–N10), and North American-cultivated American ginseng (W1–W11).

**Figure 10 foods-14-01340-f010:**
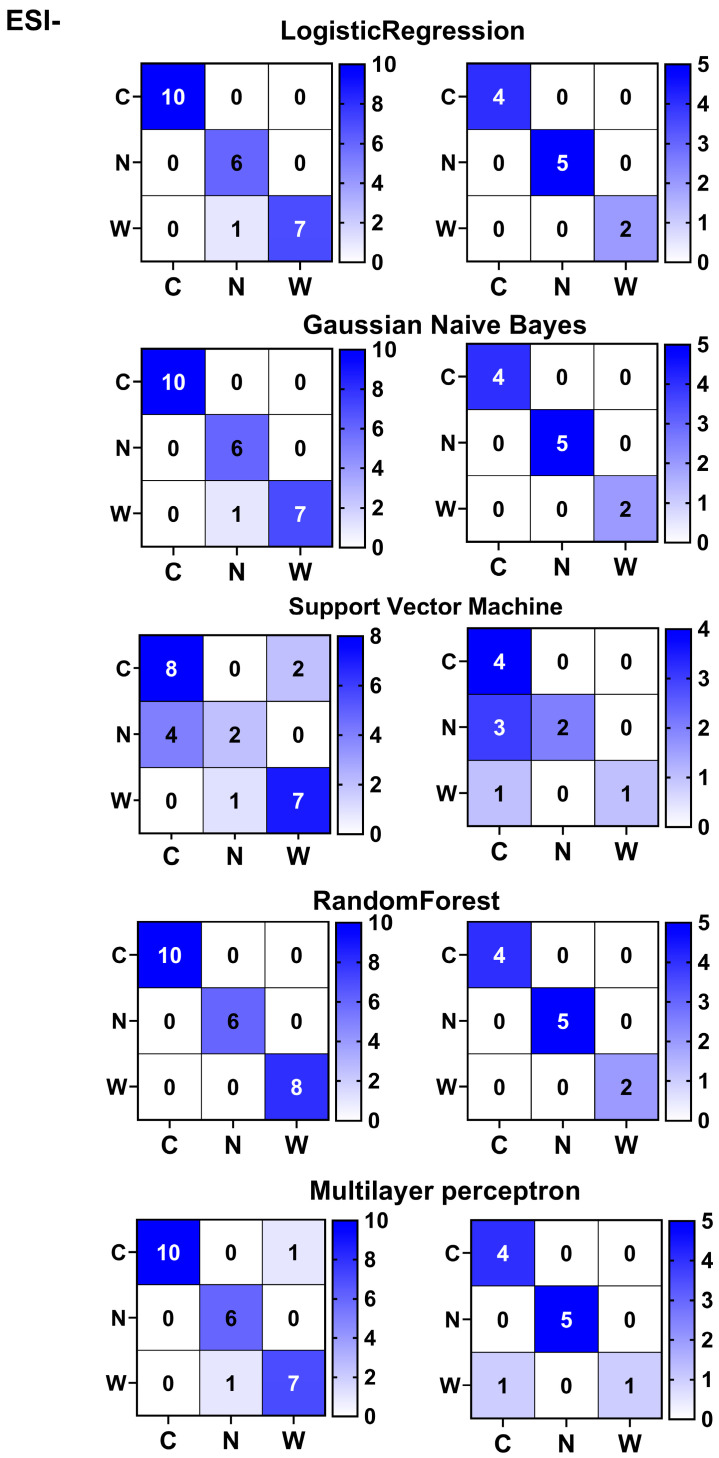
Multiclass discrimination of American ginseng in ESI- mode using machine learning models. C, N and W represent wild American ginseng (C1–C14), Chinese-cultivated American ginseng (N1–N10), and North American-cultivated American ginseng (W1–W11). Random Forest effectively scores and discriminates among three types of American ginseng in both training and test sets. Logistic Regression accurately discriminates in the training set but fails to do so in the test set. Support Vector Machine, Multilayer Perceptron, and Gaussian Naive Bayes are unable to effectively separate or discriminate among the three types of American ginseng in either training or test sets.

## Data Availability

The original contributions presented in the study are included in the article, further inquiries can be directed to the corresponding authors.

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
