# Peer review of "Cultivar Differentiation and Origin Tracing of Panax quinquefolius Using Machine Learning Model-Driven Comparative Metabolomics"

_foods, 2025, doi:10.3390/foods14081340_

Round 1

Reviewer 1 Report

Comments and Suggestions for Authors

The article Cultivar differentiation and origin tracing of American Ginsengs using Machine Learning models driven-Comparative Metabolomics driven is very well written and structured. The graphical abstract and the inserted figures greatly enriched the proposed material. I suggest small changes:

Do not use the words present in the title in the "keywords" topic

What is the objective and hypothesis of the study? Make this clear in the writing

The Results section presents discussion and this is not correct. Therefore, to prevent the text from becoming repetitive, I suggest that the authors present the Results and Discussion sections together and not separately as proposed

The last paragraph of the discussion should be inserted in the conclusion because it is a final consideration

Figures 2, 3, 7 and 8 need a better resolution so that the behavior of the data obtained can be observed

Author Response

Thank you for your letter and for the reviewers' comments concerning our manuscript (lD: foods-3300227). Those comments are all valuable and very helpful for revising and improving our paper, as well as the important guiding significance to our researches. We have studied comments carefully and have made correction which we hope meet with approval. Revised portion are marked with different colors in the paper. The main corrections in the paper and the responds to the reviewer's comments are as flowing:

Reviewer 1 Comments:

The article Cultivar differentiation and origin tracing of American Ginsengs using Machine Learning models driven-Comparative Metabolomics driven is very well written and structured. The graphical abstract and the inserted figures greatly enriched the proposed material. I suggest small changes:

Response: Thank you for your careful evaluation of this manuscript.

1) Do not use the words present in the title in the "keywords" topic

Response: Thank you. Following your suggestion, we have changed the keywords to American ginseng; metabolomics; machine learning; cultivated-origin traceability

2) What is the objective and hypothesis of the study? Make this clear in the writing

Response: Thank you for your kind suggestion, we apologized for the unclear and vague description of the objective and hypothesis and we rewrite the research

3) The Results section presents discussion and this is not correct. Therefore, to prevent the text from becoming repetitive, I suggest that the authors present the Results and Discussion sections together and not separately as proposed

Response: Thank you for your kind suggestion, we integrated the results and discussion in the manuscript

4)The last paragraph of the discussion should be inserted in the conclusion because it is a final consideration

Response: The last paragraph of the discussion has been integrated in the conclusion because it is a final consideration

5) Figures 2, 3, 7 and 8 need a better resolution so that the behavior of the data obtained can be observed

Response: Added high-resolution images.

Reviewer 2 Report

Comments and Suggestions for Authors

I reviewed the manuscript “Cultivar differentiation and origin tracing of American Ginsengs using Machine Learning models driven-Comparative Metabolomics driven” (foods-3300227). The authors’ proposed an interesting alternative to classify ginseng samples by combination of metabolomic and statistical strategies.

Some comments:

1) Please include the Figures following the text, follow the template instructions.

1) The tittle is not correct; it is mentioned American Ginsengs but the authors analyze American and Chinese Ginsengs.

2) Homogenize the terms, it is used Ginsengs and Panax quinquefolius powder indistinctly. It is not clear if samples were dried? If samples were dried, it must include the drying process.

3) Line 104. Please include the sample volumes employed for water-saturated n-butanol solution.

4) Did you analyze a unknow sample? External validation is desirable.

5) It must be included also into the discussion if the proposed strategy is robust, what about changes in metabolomic profile with storage time, geographic and temporality conditions (Year).

6) I suggest including in the Figure captions the abbreviations meaning for: C, N and W

Author Response

Thank you for your letter and for the reviewers' comments concerning our manuscript (lD: foods-3300227). Those comments are all valuable and very helpful for revising and improving our paper, as well as the important guiding significance to our researches. We have studied comments carefully and have made correction which we hope meet with approval. Revised portion are marked with different colors in the paper. The main corrections in the paper and the responds to the reviewer's comments are as flowing:

Reviewer 2 Comments:

I reviewed the manuscript “Cultivar differentiation and origin tracing of American Ginsengs using Machine Learning models driven-Comparative Metabolomics driven” (foods-3300227). The authors’ proposed an interesting alternative to classify ginseng samples by combination of metabolomic and statistical strategies.

Response: Thank you for your careful evaluation of this manuscript.

Some comments:

  • Please include the Figures following the text, follow the template instructions.

Response: Many thanks. We included the figures following the text

1) The tittle is not correct; it is mentioned American Ginsengs but the authors analyze American and Chinese Ginsengs.

Response: We apologize for the unclear and ambiguous description; we actually analyzed American cultivated- and wild- American ginseng (Panax quinquefolius) and Chinese cultivated American ginseng and now change the title to Cultivar differentiation and origin tracing of Panax quinquefolius using Machine Learning models driven-Comparative Metabolomics.

2) Homogenize the terms, it is used Ginsengs and Panax quinquefolius powder indistinctly. It is not clear if samples were dried? If samples were dried, it must include the drying process.

Response: Many thanks. We apologize for the unclear and ambiguous description; we actually analyzed American cultivated- and wild- American ginseng (Panax quinquefolius) and Chinese cultivated American ginseng. Based on your suggestion, we have removed inappropriate descriptions and supplemented the drying process of medicinal herbs.

3) Line 104. Please include the sample volumes employed for water-saturated n-butanol solution.

Response: Based on your suggestion, we supplemented the extraction volume. A quantity of 1 g precisely weighed fine powder was extracted with 50-mL water-saturated n-butanol solution in reflux extraction for 90 min.

4) Did you analyze a unknow sample? External validation is desirable.

Response: Thank you for your suggestion. We agree with your viewpoint. Due to the difficulty of collecting samples, we did our best to collect samples from 25 batches. Based on these 25 batches of samples, we conducted a machine learning exploration of the origin traceability of American ginseng. We don't have any more samples at hand, but we do offer another common,credible and convincing model evaluation, cross validation, to verify models. The detailed results are given below. The American ginseng samples were partitioned into a training set, comprising 25 samples, and a test set, comprising 10 samples, in a ratio of 5:2. A 10-fold cross-validation method was implemented to verify the stability of the algorithms. In the future, if we are able to obtain more samples of unknown western ginseng, we will perform external validation of the model.

5) It must be included also into the discussion if the proposed strategy is robust, what about changes in metabolomic profile with storage time, geographic and temporality conditions (Year).

Response: The chemical composition and quality of Panax quinquefolium are affected by a variety of factors, including geographic origins, temporality conditions and cultivation method. In this study, we focused on analyzing and comparing the effects of cultivation method and origin on the saponin composition of Panax quinquefolium, and found that the content of saponins in wild Panax quinquefolium was higher than that in cultivated Panax quinquefolium, and the saponin content of North American cultivated American ginseng was higher than that of Chinese cultivated American ginseng.

6) I suggest including in the Figure captions the abbreviations meaning for: C, N and W

Response: Added. C, N and W represent American wild ginseng, American cultivated ginseng and Chinese cultivated American ginseng, respectively.

Reviewer 3 Report

Comments and Suggestions for Authors

This paper introduces a UHPLC-Q/TOF-MS-based comparative metabolomics and machine learning strategy for the differentiation of wild and cultivated American ginseng.

 Please provide answers to the following questions:

 1. In the discussion section, the authors primarily addressed the necessity and applicability of metabolomics. However, further discussion is needed on why the three sample groups exhibit metabolomic differences and the main factors driving these differences in the context of this study’s theme.

 2. What is the rationale for employing machine learning in the classification of wild and cultivated American ginseng, despite the clear separation achieved by OPLS-DA? Explain the necessity and academic justification for using machine learning methods in this classification context.

 3. Is a sample size of 25 sufficient for conducting machine learning in this study? If so, please provide the academic justification for its adequacy. A detailed explanation is required.

 4. Additionally, multiple models were employed in the machine learning process. Please provide an analysis of why the random forest model yielded the highest performance, while other models demonstrated comparatively lower accuracy.

Author Response

Thank you for your letter and for the reviewers' comments concerning our manuscript (lD: foods-3300227). Those comments are all valuable and very helpful for revising and improving our paper, as well as the important guiding significance to our researches. We have studied comments carefully and have made correction which we hope meet with approval. Revised portion are marked with different colors in the paper. The main corrections in the paper and the responds to the reviewer's comments are as flowing:

Reviewer 3 Comments:

This paper introduces a UHPLC-Q/TOF-MS-based comparative metabolomics and machine learning strategy for the differentiation of wild and cultivated American ginseng.

 Please provide answers to the following questions:

  1. In the discussion section, the authors primarily addressed the necessity and applicability of metabolomics. However, further discussion is needed on why the three sample groups exhibit metabolomic differences and the main factors driving these differences in the context of this study’s theme.

Response: We explain and discuss that the three types of American ginseng exhibit distinct metabolic differences that are influenced by geographic origin and cultivation method. Wild American ginseng had higher levels of most saponins than cultivated American ginseng, and North American cultivated American ginseng had higher levels of most saponins than Chinese cultivated American ginseng.

  1. What is the rationale for employing machine learning in the classification of wild and cultivated American ginseng, despite the clear separation achieved by OPLS-DA? Explain the necessity and academic justification for using machine learning methods in this classification context.

Response: Thank you for your suggestion, we add relevant discussion and explanation in the manuscript. The OPLS-DA model is more suitable for comparing or classifying samples between two groups and less suitable for comparing samples among multiple groups, on the contrary, so many machine learning models are easy to compare and classify samples between among multiple groups, which helps to discriminate samples among multiple groups.

3.Is a sample size of 25 sufficient for conducting machine learning in this study? If so, please provide the academic justification for its adequacy. A detailed explanation is required.

Response: Thank you for suggesting that we provide explanations for this part of the deficiencies and limitations that we have discussed in the paper. Due to the difficulty of collecting samples, we did our best to collect samples from 25 batches. Based on these 25 batches of samples, we conducted a machine learning exploration of the origin traceability of American ginseng. In the future, if we are able to obtain more samples of unknown western ginseng, we will perform external validation of the model. In addition, at present, due to the slow accumulation of material data, the small sample size problem will still exist for a long time in is food analysis and herbal analysis, and machine learning models based on small samples are acceptable, and also food traceability models with similarly small samples (<100) are widely reported (DOI: 10.1016/j.saa.2021.120440, DOI: 10.1016/j.foodres.2022.111512,and DOI:10.1016/j.lwt.2023.115140, DOI: 10.1038/s41586-024-08328-6)

  1. Additionally, multiple models were employed in the machine learning process. Please provide an analysis of why the random forest model yielded the highest performance, while other models demonstrated comparatively lower accuracy.

Response: In our study, we compared the ability of multiple machine models to discriminate the origin of American ginseng, and found that Random Forest exhibited high accuracy for origin tracing of American ginseng by integrating multiple decision trees. In addition, Random Forest also shows strong resistance to overfitting, due to the different training data and feature sets of each tree, the model has a strong generalisation ability, which can effectively avoid the overfitting problem that is easily occurring with a single decision tree.